# Xanthone Derivatives and Their Potential Usage in the Treatment of Telangiectasia and Rosacea

**Katarzyna Brezdeń** [1,2] **and Anna M. Waszkielewicz** [1,*]

1. Department of Bioorganic Chemistry, Chair of Organic Chemistry, Faculty of Pharmacy, Jagiellonian University Medical College, Medyczna 9, 30-688 Krakow, Poland; katarzyna.brezden@uj.edu.pl
2. Doctoral School of Medical and Health Sciences, Jagiellonian University Medical College, św. Łazarza 15, 31-530 Krakow, Poland
* Correspondence: anna.waszkielewicz@uj.edu.pl

**Abstract:** Xanthone derivatives, a class of natural compounds abundantly found in plants such as mangosteen (*Garcinia mangostana*) and certain herbs, have garnered substantial interest due to their diverse pharmacological properties, including antioxidant, anti-inflammatory, and anti-cancer activities. Recent investigations have unveiled their potential as modulators of enzymatic activity, prompting exploration into their effects on hyaluronidase-mediated hyaluronic acid (HA) degradation, and their effects in topical treatment of telangiectasia and rosacea. Telangiectasia and rosacea are common dermatological conditions characterized by chronic skin inflammation, vascular abnormalities, and visible blood vessels, resulting in significant cosmetic concerns and impaired quality of life for affected individuals. This review aims to provide a comprehensive overview of the current understanding regarding the interplay between the mechanisms of action by which xanthone derivatives exert their therapeutic effects, including the inhibition of pro-inflammatory cytokines, modulation of oxidative stress pathways, and regulation of vascular endothelial growth factors. Furthermore, we will discuss the implications of harnessing xanthone derivatives as therapeutic agents for mitigating vascular dysfunction and its associated pathologies, thereby offering insights into future research directions and therapeutic strategies in the field of vascular biology.

**Keywords:** xanthone; telangiectasia; rosacea; vascular disorders; cosmetics; dermatology; blood vessels





## 1. Introduction

Vascular disorders are abnormalities in the structure and function of blood vessels. Vascular disorders of the skin are visible as dilated, sometimes cracked capillaries, veins, and arterioles but also as spots or nodules with an erythematous basis. They arise as a result of pathogenic processes, such as blood disorders in the vessels or abnormal vascular development [1]. Normally, blood vessels are formed by two mechanisms: vasculogenesis and angiogenesis. Vasculogenesis is the formation of blood vessels from the cells of the insular endothelium, leading to the blood vessel plexus. Then, the vessels are formed from the vascular endothelium in a process called angiogenesis. Under pathological conditions, various factors interfere with angiogenesis and result in excessive growth of capillaries in a tumor, a process called neoangiogenesis. Vascular endothelial growth factor (VEGF) plays a key role in the processes mentioned above [2–4]. In addition, with aging, blood vessels are degraded by the enzyme hyaluronidase, which depolymerizes hyaluronic acid and leads to a weakening of the endothelium, resulting in increased vascular permeability [5].

Chemical agents that can minimize skin lesions related to vascular abnormalities are mostly highly effective antioxidants and inhibitors of hyaluronidase. The most-used active substances are compounds from the group of flavonoids and vitamins (e.g., C, K, and B3).

The vascular structure, comprising arteries, veins, and capillaries, plays a pivotal role in maintaining physiological homeostasis within the human body. Arteries transport

oxygenated blood away from the heart to various tissues and organs, while veins return deoxygenated blood back to the heart. Capillaries, the smallest blood vessels, facilitate nutrient and gas exchange between the bloodstream and surrounding tissues. Structurally, arteries possess thicker walls rich in smooth muscle and elastic fibers, enabling them to withstand high pressure and maintain blood flow. Veins exhibit thinner walls with less muscle and more valves to prevent backflow, aiding in the return of blood to the heart. Capillaries, in contrast, consist of a single layer of endothelial cells, facilitating the efficient exchange of gases, nutrients, and waste products, such as carbon dioxide or urea. This intricate network of vessels is regulated by complex physiological mechanisms involving neural, hormonal, and local factors to ensure proper distribution of blood throughout the body. Dysfunction in vascular structure and function underlies numerous pathological conditions, including hypertension, atherosclerosis, and peripheral vascular disease, highlighting the critical importance of understanding vascular biology for the advancement of medical interventions and treatments [6]. This structure is intimately associated with the glycocalyx, a gel-like layer composed of glycoproteins and proteoglycans that coats the luminal surface of the endothelial cells lining the blood vessels. The glycocalyx serves as a dynamic interface between the bloodstream and the vessel wall, contributing significantly to vascular function and health. Within arteries and veins, the glycocalyx acts as a protective barrier, preventing the direct contact of circulating blood components with the endothelial cells and underlying tissues. This barrier function is crucial for maintaining vascular integrity and preventing inflammation and thrombosis. Moreover, the glycocalyx participates in the regulation of vascular permeability and endothelial cell signaling. It modulates the passage of nutrients, hormones, and immune cells across the vessel wall, thereby influencing tissue perfusion and immune response. Additionally, the glycocalyx plays a vital role in mechanotransduction, sensing and transducing the mechanical forces exerted by blood flow into biochemical signals that regulate vascular tone and remodeling. Furthermore, alterations in the glycocalyx have been implicated in various vascular pathologies, including endothelial dysfunction, atherosclerosis, and diabetic vascular complications. Damage to the glycocalyx, often induced by oxidative stress, inflammation, or hyperglycemia, compromises its protective and regulatory functions, contributing to vascular injury and disease progression [7].

Hyaluronic acid is a large, negatively charged glycosaminoglycan that contributes to the hydration, lubrication, and structural integrity of the glycocalyx. Through its ability to interact with water molecules and form a hydrated gel-like matrix, hyaluronic acid helps maintain the thickness and viscoelastic properties of the glycocalyx, which is crucial for its barrier function and mechanotransduction [8].

Hyaluronidase is an enzyme of the hydrolase class that depolymerizes the hyaluronic acid located in blood vessel walls (Scheme 1). Hyaluronidase is an endoglycosidase that breaks down hyaluronic acid into monosaccharides by cleaving its glycosidic bonds. Additionally, to some extent, it also breaks down other acid mucopolysaccharides in the connective tissue [9]. It is present both in organs and body fluids (e.g., tears and blood). There are six types of hyaluronidases in the human body: hyaluronidases 1–4, encoded by HYAL genes 1–4, PH-20 (SPAM-1), and HYALP1. Hyaluronidase 1 is present in human organs and acts as the major hyaluronidase in plasma, being activated at acidic pH levels. Hyaluronidase 2, on the other hand, has weaker enzymatic activity than hyaluronidase 1 and degrades only high molecular weight hyaluronic acid. It is mostly used in aesthetic medicine to dissolve tissue fillers based on high molecular weight hyaluronic acid. The role of hyaluronidase 3 remains unknown, as it is found only in the nucleus and bone marrow. Testicular hyaluronidase PH-20 is found on the surface of human sperm and the inner acrosomal membrane and serves to degrade hyaluronic acid in the egg cell during fertilization [10].

**Hyaluronic acid (HA)**

**Scheme 1.** Mechanism of action of hyaluronidase. Adapted from Lee A. et al. [11] and Ponnuraj K. et al. [12].

Recent studies have shed light on the intricate interplay between hyaluronidase, hyaluronic acid, and the vascular glycocalyx, providing valuable insights into their roles in vascular physiology and pathology. For instance, research by Tarbell and Cancel found that enzymatic degradation of hyaluronic acid by hyaluronidase disrupts the glycocalyx structure, leading to increased vascular permeability and endothelial dysfunction [13]. Moreover, work by Florian et al. demonstrated that hyaluronidase-mediated degradation of hyaluronic acid fragments can stimulate inflammatory responses and promote atherosclerosis progression in animal models [14].

Furthermore, clinical studies by Becker et al. [15] have highlighted the association between elevated hyaluronidase activity and impaired glycocalyx function in patients with cardiovascular diseases, underscoring the potential clinical relevance of targeting hyaluronidase for vascular protection. Additionally, investigations by Slevin et al. [16] have identified dysregulated hyaluronidase expression in diabetic vasculopathy, suggesting a link between altered hyaluronic acid metabolism and diabetic vascular complications [15].

These findings collectively emphasize the importance of understanding the dynamic interactions between hyaluronidase, hyaluronic acid, and the glycocalyx in vascular health and disease. Future research efforts aimed at elucidating the molecular mechanisms underlying these interactions may pave the way for the development of novel therapeutic approaches for vascular disorders [16].

## 2. Xanthone Derivatives and Their Vascular Activity

Xanthone (9*H*-xanthen-9-one, dibenzo-γ-pyrone) derivatives are oxygen-containing heterocycles (Figure 1). Many oxygenated heterocycles possess pharmacological activities, and the xanthone class is not an exception [17]. At present, nearly 1000 naturally occurring xanthone derivatives are known. Each has different substituents at different positions,

ultimately leading to a large variety of pharmacological activities [18,19]. The multitude of biological activities found for xanthone derivatives include α-glucosidase inhibition, anti-cancer, anti-Alzheimer, anticonvulsant, anxiolytic, antidepressant, analgesic, antibacterial, antioxidant, and anti-inflammatory activities [20–25].

**Figure 1.** Structure of xanthone (dibenzo-γ-pyrone).

*2.1. Sealing Blood Vessel Activity*

One of the intriguing therapeutic potentials of xanthone derivatives lies in their ability to modulate vascular function, particularly when connected with sealing blood vessels. The integrity of blood vessels is crucial for maintaining proper cardiovascular health, as disruptions in their structure or function can lead to various pathological conditions, including hemorrhaging, thrombosis, and atherosclerosis, and connections with the skin, including telangiectasia and rosacea.

Research has increasingly elucidated the mechanisms by which various xanthone derivatives exert their vascular sealing effects. These mechanisms often involve interactions with key molecular targets involved in vascular homeostasis, such as endothelial cells, smooth muscle cells, and inflammatory mediators. For instance, xanthone derivatives have been shown to enhance endothelial barrier function by regulating the expression of tight junction proteins and inhibiting the production of pro-inflammatory cytokines [26].

2.1.1. Antiangiogenic Activity

Angiogenesis refers to the formation of new blood vessels from existing ones, and inhibiting this process holds promise in combating cancer. Tumor angiogenesis, a complex series of events involving endothelial cell activation, invasion, migration, proliferation, and the formation of capillary networks, is crucial for cancer progression and metastasis. Without the formation of new blood vessels, tumors typically remain small and are less likely to metastasize. Consequently, targeting angiogenesis is a significant strategy in cancer treatment with regard to skin lesions as well.

In one of the studies conducted by Tao Chan et al., a few gambogic acid derivatives were obtained and evaluated in terms of antiangiogenic activity. Four derivatives effectively inhibited the development of new segmental blood vessels in zebrafish assays and showed lower toxicity to zebrafish than gambogic acid (GA) (Figure 2). They also demonstrated stronger inhibitory effects on the migration and tube formation of human umbilical vein endothelial cells (HUVECs) in vitro compared with GA [27].

Another xanthone derivative, α-mangostin (Figure 3), exhibits significant inhibitory effects on various aspects of angiogenesis and vascular permeability in retinal endothelial cells (RECs). Specifically, it suppresses VEGF-induced permeability increases, proliferation, migration, and tube formation, as well as vascular sprouting in the aortic ring assay. This compound also hampers the phosphorylation of VEGF receptor 2 (VEGFR2) as well as extra-cellular signal-regulated kinase 1 and 2-mitogen-activated protein kinase (ERK1/2-MAPK) induced by VEGF. Furthermore, α-mangostin has been found to inhibit reactive oxygen species (ROS) formation induced by hypoxia treatment in RECs. This suggests its potential in mitigating oxidative stress associated with hypoxia and preventing neovascularization in the retina. The inhibition of VEGF-induced angiogenic responses by α-mangostin is associated with its ability to block VEGFR2 and ERK1/2-MAPK activation. These findings underscore the antioxidant and anti-angiogenic properties of α-mangostin, indicating its potential as a natural therapeutic agent for reducing oxidative stress and preventing retinal neovascularization. Moreover, α-mangostin's selective inhibition of specific signaling path-

ways like VEGFR2 and ERK1/2-MAPK without affecting Akt and p38 phosphorylation provides insights into its mechanism of action in regulating angiogenesis and vascular permeability in microvascular endothelial cells. It was successfully demonstrated that α-mangostin possesses both antioxidant and anti-angiogenic properties when applied to RECs. Specifically, our findings reveal that α-mangostin effectively reduces the formation of ROS triggered by hypoxia treatment in RECs, α-mangostin inhibits various angiogenic responses induced by VEGF, such as enhanced REC permeability, proliferation, migration, tube formation, and vascular sprouting in the aortic ring assay, and α-mangostin attenuates the phosphorylation of the VEGFR2 and ERK1/2-MAPK pathways in RECs stimulated by VEGF [28].

**Figure 2.** Structure of gambogic acid: (Z)-4-((1S,3aR,5S,11R,14aS)-8-hydroxy-2,2,11-trimethyl-13-(3-methylbut-2-en-1-yl)-11-(4-methylpent-3-en-1-yl)-4,7-dioxo-1,2,5,7-tetrahydro-11H-1,5-methano-furo[3,2-g]pyrano[3,2-b]xanthen-3a(4H)-yl)-2-methylbut-2-enoic acid).

**Figure 3.** Structure of α-mangostin (1,3,6-trihydroxy-7-methoxy-2,8-bis(3-methylbut-2-en-1-yl)-xanthone).

### 2.1.2. Endothelial and Mitochondrial Dysfunction

Mitochondrial dysfunction arises from disruptions in mitochondrial HK-II activity. Mangiferin (Figure 4) effectively preserves mitochondrial HK-II, thereby safeguarding overall mitochondrial function. In endothelial cells, palmitic acid (PA) stimulation induces oxidative stress, as evidenced by heightened ROS production, which is mitigated by mangiferin at concentrations of 1.0 and 10 μM. Considering the vital role of mitochondria in energy production and cell survival, mangiferin's ability to promote HK-II binding to mitochondria through protein kinase B (Akt) activation becomes crucial. This mechanism not only protects mitochondrial function in vessel endothelial cells but also suggests a potential therapeutic avenue for mitigating endothelial dysfunction by preserving mitochondrial HK-II via pharmacological Akt activation. Akt activation stimulates downstream responses and promotes cell survival by mediating the cellular growth factors and blocking apoptosis through the inactivation of pro-apoptotic proteins. Consequently, mangiferin reduces caspase-3 expression and safeguards a vessel's endothelial integrity by modulating ROS

generation and enhancing NO production, as demonstrated in vivo. This study further underscores the significance of mitochondrial HK-II for maintaining mitochondrial functional integrity in vessel endothelial cells. Through Akt activation, mangiferin fosters HK-II binding to mitochondria, thus ameliorating mitochondrial dysregulation and promoting endothelial cell survival [29].

**Figure 4.** Structure of mangiferin: 1,3,6,7-tetrahydroxy-2-((2S,3R,4R,5S,6R)-3,4,5-trihydroxy-6-(hydroxymethyl)tetrahydro-2H-pyran-2-yl)-xanthone.

In native, healthy endothelial cells, a number of stimuli (e.g., serotonin from aggregating platelets, sphingosine 1-phosphate, and thrombin) activate eNOS, causing the release of NO, which relaxes the vascular smooth muscle that surrounds them. NO, in synergy with prostacyclin, further inhibits platelet aggregation [30].

It has been reported that endothelial dysfunction is associated with cell apoptosis, migration, and then inflammation [31].

It has been shown that endothelium-dependent vasodilation is attenuated in many risk factors of atherosclerosis, such as hypercholesterolemia, hypertension, and diabetes mellitus, and endothelial dysfunction is recognized as an early event in the pathogenesis of atherosclerosis which can be visible on the skin, such as xanthomas. Xanthomas are yellowish or orange bumps or nodules that develop under the skin due to the deposition of cholesterol-rich material [32]. Nitric oxide (NO), synthesized from *L*-arginine by NO synthase (NOS) in endothelial cells, has been thought to play a key role in the maintenance of vascular tone and structure. NO possesses complex cardiovascular actions such as protection of endothelial cells and decreasing the endothelial adhesiveness and inhibition of the adhesion of monocytes to endothelial cells, and it is generally described as an "endogenous antiatherosclerotic molecule". Recently, it has been found that *L*-arginine analogues such as asymmetric dimethylarginine (ADMA), which is present in the blood of both humans and animals, can inhibit NOS in vivo and in vitro. ADMA, depending on its concentration, has been shown to inhibit vasodilator responses to acetylcholine in isolated aortic rings, upregulate the expression of monocyte chemoattractant protein-1 (MCP-1), and enhance the adhesion of monocytes in cultured endothelial cells. There is growing evidence that endothelial dysfunction in some cardiovascular diseases, such as hypercholesterolemia, heart failure, and hypertension, is associated with elevated ADMA levels, and its levels could predict endothelial dysfunction [33].

Vasodilator responses to acetylcholine in rings of the isolated thoracic aorta have been shown to be impaired in the presence of lysophosphatidylcholine (LPC), a major component of ox-LDL. Daviditin A (Figure 5) significantly attenuates the inhibition of endothelium-dependent relaxation by LPC. Moreover, in cultured endothelial cells, xanthone derivatives inhibited the increase in the release of LDH, the upregulation of MCP-1 expression, and the enhancement of monocyte adhesion concomitantly with a reduction in ADMA levels. These findings suggest that xanthone derivatives protect against endothelial damage induced by high lipid levels, and their protective effect on the endothelium is related to a reduction in the ADMA concentration [34].

**Figure 5.** Davitin A (1,8-dihydroxy-2,5,6-trimethoxyxanthone) structure.

### 2.1.3. Inhibition of Hyaluronidase

The inhibition of hyaluronidase by xanthone derivatives is a novel concept, being related to permeability through the blood vessels. Only one study was found in which all tested compounds were inactive against hyaluronidase. The structures tested were xanthone derivatives with only methoxy, hydroxy, or methylbromo moieties [35]. On the other hand, it was found that $\gamma$-mangostin (Figure 6) showed potent values of inhibition of hyaluronidase (IC$_{50}$ 23.85 $\mu g \, mL^{-1}$) [36].

**Figure 6.** Structure of $\gamma$-mangostin (1,3,6,7-tetrahydroxy-2,8-bis(3-methylbut-2-en-1-yl)-xanthone).

### 2.1.4. Diabetic Vascular Complications

Telangiectasia can be also caused by diabetic vascular changes, often followed by microangiopathy.

Diabetic vascular complications represent a significant cause of mortality and morbidity in individuals with diabetes, particularly those with type 2 diabetes. These complications are marked by endothelial dysfunction, with impaired NO production playing a central role. NO is crucially synthesized in vascular endothelial cells through the activation of endothelial nitric oxide synthase (eNOS). However, in diabetes, eNOS activity may be compromised due to the accumulation of ceramide. Recent studies have highlighted the potential anti-diabetic properties of $\alpha$-mangostin [37].

Research demonstrates that $\alpha$-mangostin ameliorates impaired endothelium-dependent vasodilation (EDV) in diabetic animals by modulating the aSMase/ceramide pathway and enhancing the eNOS/NO pathway in aortic tissues. Additionally, $\alpha$-mangostin inhibited the elevated aSMase/ceramide pathway and restored EDV in isolated mouse aortas exposed to high-glucose conditions. Moreover, $\alpha$-mangostin increased eNOS phosphorylation and NO production in aortic tissues treated with high glucose. Furthermore, $\alpha$-mangostin normalized the activation of the aSMase/ceramide pathway and improved the eNOS/NO pathway in endothelial cells exposed to high glucose concentrations. Overall, studies revealed that $\alpha$-mangostin regulates the eNOS/NO pathway and enhances EDV in diabetic mice by inhibiting aSMase activity and reducing endogenous ceramide accumulation. These results provide novel insights into the therapeutic potential of $\alpha$-mangostin in alleviating endothelial dysfunction associated with diabetes. Additionally, these findings expand upon previous research demonstrating the beneficial effects of inhibiting ceramide synthesis on arterial function in diabetic animals and reversing NO levels in endothelial cells exposed to high glucose [38,39].

## 2.2. Anti-Inflammatory Mechanisms of Xanthone Derivatives

The anti-inflammatory effects of xanthone derivatives are attributed to their ability to modulate key molecular targets involved in the inflammatory process. These targets include pro-inflammatory enzymes such as cyclooxygenase (COX) and lipoxygenase (LOX), as well as inflammatory mediators like cytokines (e.g., interleukins IL and tumor necrosis factor-alpha TNF-$\alpha$), reactive oxygen species (ROS), and NO. The anti-inflammatory effects were shown based on a study where a few xanthone derivatives were isolated from the stem bark of *Garcinia delpyana*. The following compounds were isolated: two new ones, delpyxanthone A and delpyxanthone B, together with four known ones, gerontoxanthone I, $\alpha$-mangostin, cowanin, and cowanol (Figure 7). Additionally, xanthone derivatives have been shown to influence signaling pathways such as nuclear factor-kappa B (NF-$\kappa$B) and MAPKs, which play pivotal roles in orchestrating inflammatory responses. Furthermore, the antioxidant properties of xanthone derivatives contribute to their anti-inflammatory effects by neutralizing ROS and mitigating oxidative stress, a common feature of inflammatory conditions. Additionally, xanthone derivatives may modulate immune responses by regulating the activity of immune cells involved in inflammation, further enhancing their anti-inflammatory actions. Xanthones have been shown to influence the production and activity of cytokines such as interleukins (ILs) and tumor necrosis factor alpha (TNF-$\alpha$). By modulating cytokine production, xanthones can affect the balance between pro-inflammatory and anti-inflammatory responses, thereby regulating immune function. These compounds can also affect the activities of macro macrophages, T cells, and dendritic cells, leading to pausing the activation of pro-inflammatory immune cells or enhancing the function of regulatory immune cells and thereby exerting anti-inflammatory effects. [40].

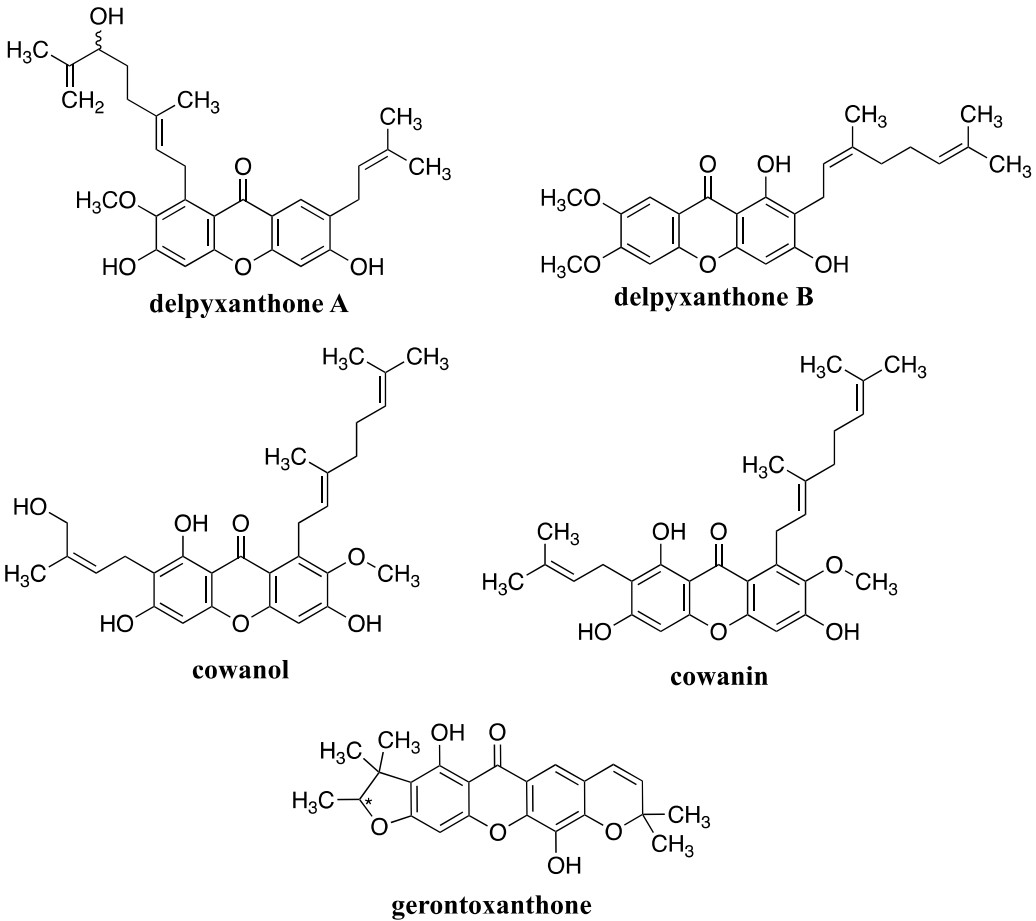

**Figure 7.** Structures of xanthone derivatives isolated from *Garcinia delpyana*.

Investigations on the potential of *G. mangostana* L. pericarps ethanol extracts for acne treatment revealed its anti-inflammatory effects, suppressing the production of the pro-inflammatory cytokine TNF-α. Some research suggests that xanthone derivatives possess antibacterial properties that could be beneficial for treating conditions like acne. Specifically, studies have shown that xanthone derivatives exhibit antibacterial activity against bacteria such as *Propionibacterium acnes* and *Staphylococcus epidermidis*, which are commonly associated with acne [41]. The formation of acne is also associated with one of the stages of rosacea.

## 2.3. Antioxidant Activity

Several studies have investigated the antioxidant activity of xanthone derivatives and have found promising results:

- Free radical scavenging: Both 1,3,7-hydroxyxanthone and other hydroxy xanthone derivatives were shown to effectively scavenge free radicals, thereby reducing oxidative stress and preventing cellular damage as well as protecting against ultraviolet (UV) radiation and pollution [42].
- Anti-inflammatory effects: Inflammation is closely linked to oxidative stress, and xanthone derivatives like α-mangostin, cowanol, and cowanin have demonstrated anti-inflammatory properties, which may contribute to their overall antioxidant activity.
- Neuroprotective effects: Some studies suggest that some xanthone derivatives, like mangiferin, may have neuroprotective effects by scavenging free radicals and reducing oxidative damage in the brain, which could potentially help with the prevention or treatment of neurodegenerative diseases such as Alzheimer's disease and Parkinson's disease [43–46].

M. Abate et al. [47] provided novel scientific evidence demonstrating the protective effects of mangostanin (Figure 8), a xanthone derivative isolated from *Garcinia mangostana* fruit, on epidermal keratinocytes against oxidative stress-induced damage. Pretreatment with mangostanin significantly attenuated $H_2O_2$-induced cytotoxicity and ROS production. Mechanistically, mangostanin inhibited the activation of key signaling pathways, including p53, p38 MAPK, ERK, and AKT, as well as the cleavage of Caspase-9 and Caspase-3 while preserving critical cell survival signals such as EGFR and signal transducer and activator of transcription 3 (STAT-3). These findings suggest a potential pharmaceutical application for mangostanin in skin protection and aging. However, further in vivo and clinical studies are warranted to validate its efficacy and safety for potential cosmeceutical, pharmacological, or nutraceutical formulations [47].

**Figure 8.** Structure of mangostanin ((R)-4,8-dihydroxy-2-(2-hydroxypropan-2-yl)-7-methoxy-6-(3-methylbut-2-en-1-yl)-2,3-dihydro-5H-furo[3,2-b]xanthen-5-one).

As mentioned above, xanthone derivatives demonstrate potent activity toward targets and pathways associated with vascular disorders. This activity warrants further investigation, specifically for potential usage of xanthone derivatives as active pharmaceutical ingredients in topical formulations.

### 3. Skin Lesions Related to Vascular Disorders

Erythema is a red, delimited lesion caused by long-term dilatation of capillaries and plasma extravasation caused by inflammatory infiltrates fading under pressure [48,49].

Telangiectasias are enlarged superficial capillaries of the sub-papillary layer of the skin which are translucent through the epidermis. They have a blue, brown, or light red color with a diameter of less than 1 mm. They occur individually or in clusters, probably arising as a result of the backflow of blood from abnormally functioning veins, local increase in venous pressure, and unstable walls of small blood vessels in the skin. The initial clinical symptom of telangiectasia is a short-term erythema that becomes permanent over time. Their occurrence is characteristic of people with couperose skin, also called *prerosacea* in the literature. Telangiectasia is also one of the first symptoms of emerging venous insufficiency of the lower extremities [1,50].

Rosacea is an erythematous dermatosis of a diverse classified etiology, belonging to the vasomotor neurosis of the skin. Clinical symptoms include erythematous, popular, and pustular lesions on seborrheic skin with vascular disorders. In the initial stage of the disease, known as *prerosacea*, we observe the occurrence of paroxysmal erythema, which remains fixed over time, along with the presence of telangiectasia. There are four subtypes of rosacea: erythematotelangiectatic (ETR), papulopustular (PPR), phymatous (PRY), and ocular. In the ETR subtype, erythema and teleangiectasis are observed. On the other hand, in the PPR subtype, papules also appear or occur. The etiology of rosacea seems quite complex, although there are three main hypotheses. The first is causing inflammation in blood vessels due to photoaging. The mechanism of photoaging, in which inflammatory factors are released, such as matrix metalloproteinases, weakens the walls of blood vessels, leading to the formation of telangiectasia. Moreover, the presence of polymorphism in the VEGF gene, which is a strong mediator of vascular permeability and inflammation, has been demonstrated in people with rosacea [51]. Another hypothesis explaining the etiology of rosacea is skin infection by the Gram-negative bacterium *Helicobacter pylori*, which lives on the surface of gastric epithelial cells. This is the least likely hypothesis. A meta-analysis [52] of 42 articles showed a weak association between rosacea and *H. pylori* infection as well as a weak effect of anti-*Helicobacter* therapy on rosacea. The last hypothesis (the most likely one) explaining the etiology of dermatosis is skin infection with the *Demodex folliculorum* parasite, called *Demodex*, inhabiting the sebaceous glands of the skin and eyelash hair capsules [11–14].

Varicose veins are permanent, superficial vein extensions that protrude above the skin's surface. They are classified as the second stage of chronic venous insufficiency. When considering contributory factors, varicose veins are categorized into primary and secondary types. Primary varicose veins are the term used to describe varicose veins of an unknown etiological factor occurring in the normal condition of deep veins. On the other hand, secondary varicose veins are vascular changes that are a complication of a history of thrombosis, insufficiency of deep veins, piercing veins, or arteriovenous fistulas. An important aspect of the pathogenesis of varicose veins is the change in the ratio of type I collagen (which strengthens the vascular walls) to type III collagen (which is responsible for tissue stretching), resulting from disturbance of the post-translational or degradation processes. This change reduces the flexibility of the walls. Varicose veins may be accompanied by symptoms such as pain, heaviness in the legs, swelling, and tingling, but bleeding may also occur after minor injuries [1,14,15].

Angiomas are vascular changes caused by excessive growth of the endothelial cells of blood vessels, with the most common ones being benign tumors of developmental age. They are characterized by high metabolic activity with an increased exchange of endothelial cells, mast cells, fibroblasts, and macrophages during the period of proliferation. They manifest as erythematosus vascular lesions which are clearly delimited with increased cohesiveness and usually raised above the skin's surface, concerning the dermis and subcutaneous tissue. The clinical appearance of hemangiomas includes single lesions or

clusters of erythematous changes on an unchanged substrate, often colloquially referred to as traces of strawberries [53,54].

Vascular malformations are a group of non-neoplastic, tumour-like vascular changes resulting from disturbed vascular tissue morphogenesis. They are most often present from birth but may remain invisible and reveal themselves later in life. Vascular malformations have been grouped based on a common embryological origin, consisting of a lining made of a single endothelial cell. Vascular defects are believed to result from developmental errors during embryogenesis, which include abnormal signaling processes controlling apoptosis, maturation, and vascular cell growth. These errors result in the cells of the vascular plexus retaining some degree of differentiation [55,56]. There are four main categories of vascular malformations based on their flow characteristics:

- Slow flow: capillary malformation, venous malformation, and lymphatic malformation;
- Rapid flow: arteriovenous malformation.

They appear on the skin as extensive, clearly demarcated erythematous spots [57,58].

## 4. Epidemiology

Vascular lesions, or more precisely skin manifestations of circulatory disorders, are still one of the main therapeutic but also aesthetic problems.

The most common vascular lesions in the population include telangiectasias, varicose veins, rosacea, hemangiomas, and vascular malformations. Hemangiomas occur in approximately 4–5% of newborns, especially Caucasians, and in up to 30% of children born prematurely. Additionally, they occur 3–9 times more often in girls than in boys. Vascular malformations occur much less frequently but with the same gender frequency [59,60].

The prevalence of venous problems of the lower extremities varies considerably with the latitude, with the highest rates reported in Western countries. Varicose veins, estimated to be more common than chronic venous insufficiency, range from <1% to 73% in women and from 2% to 56% in men. The main risk factors are age, gender, standing work, pregnancy, and a genetic family history of varicose veins [61,62].

The epidemiology of telangiectasia was investigated by conducting a large cohort study, determining the prevalence from facial photographs of more than 2000 participants from Northern Europe. This group was predominantly female (58.8%) with a median age of 66.9. The study showed that the risk factors for telangiectasia are older age, being female, smoking, particularly active and prolonged smoking, a fair complexion, and high susceptibility to sunburn (phototypes I and II on the Fitzpatrick scale). The effect causing the greatest increase in the occurrence of telangiectasias was long-term smoking, increasing the risk by 38.4%. The authors emphasized that facial erythema and telangiectasias are often associated with the erythematous teleangiectatic subtype of rosacea [63].

Rosacea is a condition that affects as much as 2–22% of the Caucasian population. It appears that the highest risk of rosacea is found in Caucasians with fair, sun-sensitive skin, classified as skin phototypes I and II according to the Fitzpatrick scale. However, a recent study reported that rosacea also occurs in phototypes III and VI, although papulopustular lesions are more frequently observed. The reason for this is the masking of erythema by the pigment of a darker phototype of skin. A prospective study conducted in Germany in 2016 found an overall 12% prevalence of rosacea, with the erythematous-vascular-pustular and papulopustular subtypes accounting for 9% and 3%, respectively. Rosacea occurs with a strong predominance in women and is usually diagnosed after the age of 30 [63–66].

The activities mentioned above for the xanthone derivatives are connected with the pathogenesis and formation of teleangiectasia and the early stages of rosacea, which reveals a need for further research about topical formulations containing synthetic xanthone derivatives.

## 5. Discussion

The exploration of xanthone derivatives as potential agents for the treatment of telangiectasia and rosacea offers a promising avenue for addressing the unmet needs of patients

with these dermatological conditions. Through their multifaceted pharmacological properties, including anti-inflammatory, antioxidant, and antimicrobial effects, xanthone derivatives demonstrate the potential to modulate the underlying pathophysiological mechanisms driving vascular abnormalities and cutaneous inflammation.

Despite the limited number of studies investigating the use of xanthone derivatives specifically for telangiectasia and rosacea, the available evidence suggests encouraging outcomes in terms of efficacy and safety. However, further research is imperative to fully elucidate the therapeutic potential of these compounds and optimize their clinical utility.

Future studies should focus on elucidating the precise mechanisms of action of xanthone derivatives in the context of telangiectasia and rosacea, as well as exploring their synergistic effects with existing treatment modalities. Additionally, efforts to identify the most effective formulations, dosing regimens, and delivery systems for topical application are warranted to maximize therapeutic outcomes and patient compliance, especially for evaluating their bioavailability.

Furthermore, rigorous clinical trials are needed to validate the findings from preclinical studies and establish the long-term safety profile of xanthone derivatives in the treatment of telangiectasia and rosacea. Close monitoring of any adverse effects and potential drug interactions will be essential to ensure the safety and tolerability of these compounds in clinical practice.

In conclusion, the growing body of evidence supporting the therapeutic potential of xanthone derivatives underscores the importance of continued research in this area. With further investigation and refinement, xanthone-based formulations may emerge as valuable additions to the therapeutic armamentarium for telangiectasia and rosacea, offering new hope for patients burdened by these challenging dermatological conditions.

**Funding:** This research was funded by Jagiellonian University Medical College, U1C/W42/NO/28.16.

**Data Availability Statement:** The data are contained within this article.

**Conflicts of Interest:** The authors declare no conflicts of interest.

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
