# Peer review of "Xanthone Derivatives and Their Potential Usage in the Treatment of Telangiectasia and Rosacea"

_applsci, doi:10.3390/app14104037_

Round 1

Reviewer 1 Report

Comments and Suggestions for Authors

Major Points:

1.    Figure problem: There are too many errors in the figures of xanthone structures. For example, figure 2. The structure of gambogic acid is wrong and looks weird. Both figure 3 and 7 are the structure of α-mangostin. Why? For figure 7, the numbers has been inserted into the figure. And for figure 8, the chirality of gerontoxanthone has not been labeled.

2.    Logical problem: Although the title of this article is xanthone derivatives for potential use in topical treatment of telangiectasia and rosacea, it reads like the integration of two independent reviews, one is on the development of telangiectasia and rosacea, the other part is on the medicinal use of xanthone derivatives. The introduction in this work is too in-detail without enough consistency. Besides, there are too many paragraphs in this article, which are not necessary.

3.    Writing problem: Authors have used a lot of “on the other hand” in the manuscript, however, the content does not express the adversative relationship. Additionally, for example, line 170-171.” Angiomas are vascular changes, the most common benign tumors of developmental age. Caused by excessive growth of endothelial cells of blood vessels.” This should be one complete sentence instead of two. Therefore, this manuscript should be polished before submission.

4.    Format problem: The format in the reference part is also problematic. For example, some references have listed their publication date, while some did not (only month/year).

Comments on the Quality of English Language

1.    Authors have used a lot of “on the other hand” in the manuscript, however, the content does not express the adversative relationship. Additionally, for example, line 170-171.” Angiomas are vascular changes, the most common benign tumors of developmental age. Caused by excessive growth of endothelial cells of blood vessels.” This should be one complete sentence instead of two. Therefore, this manuscript should be polished before submission.

Reviewer 2 Report

Comments and Suggestions for Authors

The authors provide an overview of various xanthone activities in treating several inflammatory skin and vascular ailments. The subject treatment is rather superficial overall, as in ‘this xanthone was shown to do this’ with no critical analysis or underlying mechanistic insight. That is not entirely the fault of the authors, but rather, the limited scientific data associated with the studies they emphasize. The review title and the Conclusion section are misleading, since much of the review emphasizes xanthone treatments of vascular afflictions other than of telangiectasia and rosacea. In addition, there is no discussion of topical administration of formulations including xanthones as the API until the Conclusion section. Thus, neither the title nor the Conclusion fit the review and should be modified accordingly.

Much of the review is written in idiomatic English, but some sections are not. Examples include the choice of words, phrasing, ambiguous pronouns, and incomplete sentences in the following lines: 36-43, 62 (which structure?), 196, 236-238, 252 (nonsensical as written), 351-353, 395-411 (poorly written and edited), 416-423, and 427-434.

Other major issues include:

·         The structure of gambogic acid in Figure 2 is incorrect. Gambogic acid is an enoic acid.

·         The hyaluronic acid polymer in Scheme 1 is depicted incorrectly. It should be drawn as a disaccharide repeat unit (rather than as a tetrasaccharide) with connections at C1 and C4. The stereodefined chiral centers at C3 neighboring the acetamido substituents are also missing.

·         Are the references in lines 112-115 correct?

·         Sections of the document are inadequately referenced. A reference should be added to line 371, and references must be added to most sentences throughout section 4.1.3 (lines 397-418). This section is very poorly referenced.

·         A better description of immune modulation is required in lines 409-411.

·         What do the ranges in lines 200-201 reflect? “<1 to 73%” of what? ‘None to most’ of something undefined is not useful to the reader. The qualifications of those ranges are necessary for the ranges to have any meaning.

·         The neuroprotective effects espoused in lines 438-439 are highly speculative and should be supported with better evidence.

Overall, this is not a bad review, and it should appeal at its heart to many readers. However, the title and the Conclusion do not match the emphasized content and are therefore misleading. The topical component should be removed entirely unless it is described more completely within the document. Some sections are wonderfully clear and nicely referenced, while those noted above require substantial modification, because they are insufficiently cogent and inadequately referenced, at present. I would ultimately recommend publication; however, not until the corresponding changes to the title, Conclusion, and indicated artwork and paragraphs are made by the authors.

Comments on the Quality of English Language

Please see my notes in the Comments and Suggestions for the Authors.

Reviewer 3 Report

Comments and Suggestions for Authors

Dear Authors,

The manuscript is interesting and give us a well-written review about xanthones derivatives. However, in order to improve the quality of the text I recommend to change the order of the sections. 

It's essential to start the review by giving the explanation about which are the xanthones derivates, where are they obtained. Then, you can continue with all the information about its mechanism of action in skin and in vascular disorders. At the end of the text, you could explain tha pathogenesis of rosacea and the potential use of xanthone derivatives in the treatment of this diseases.

I hope these suggestions could help you to improve the manuscript.

Reviewer 4 Report

Comments and Suggestions for Authors

Enjoyed reviewing the manuscript, very nice work.  I have made few very minor comments/suggestions for your consideration. 

Good Luck

Round 2

Reviewer 1 Report

Comments and Suggestions for Authors

Thank you for the revision. I recommend this work to be accepted by Applied Science.

Author Response

Dear Reviewer,

Thank you so much once again for your time and all the notes.

Kind regards,

Authors

Reviewer 2 Report

Comments and Suggestions for Authors

The authors made several changes to address the issues associated with the initial submission, but many identified deficiencies remain in the edited version. To assist, I will help with phrasing of a few passages to offer corrections and, by my interpretation, clearer delivery of each corresponding point. The authors are welcome to incorporate my suggested changes exactly as written or modify these sections again to provide their own clarifications. However, other sections for which I do not offer corrections also require significant editing for English grammar, just as in the initial submission.   

·         Lines 36-43 (rewritten with corrections):Then, the vessels are formed from the vascular endothelium in a process called angiogenesis. Under pathological conditions, various factors interfere with angiogenesis which result in excessive growth of capillaries in the tumour—a process called neoangiogenesis. Vascular endothelial growth factor (VEGF) plays a key role in the processes mentioned above [2–4]. In addition, with aging, blood vessels are degraded by the enzyme hyaluronidase, which depolymerizes hyaluronic acid and leads to a weakening of the endothelium resulting in increased vascular permeability [5].

·         All structures in Scheme 1 are still incorrectly drawn. The hyaluronic acid must be drawn as a polymer connected through the 4’-OH of the glucuronic acid portion to the anomeric position of the N-acetylglucosamine portion. The N-acetylglucosamine portion of the hyaluronic acid is also missing an equatorial 3’-OH. In addition, the wedge is still missing from the 3’ position of the N-acetylglucosamine in one HYAL product, while both HYAL products must remain polymers at one position (i.e., the HYAL makes those terminal positions of shortened polymers and not free disaccharides, as shown). This scheme should not be published as drawn because it is not chemically accurate.

·         Line 140 (rewritten with corrections): thrombosis, atherosclerosis, and with connections to the skin including telangiectasia and rosacea.

·         Lines 152-153 (rewritten with correction): without the formation of new blood vessels, tumors typically remain small and are less likely to metastasize. Consequently, targeting angiogenesis is a significant

·         Although the structure of gambogic acid in Figure 2 was corrected, it is not drawn in a manner that makes it interpretable (i.e., too many overlapping atom labels and bonds.) I recommend the authors seek assistance with the graphic or remove it from the document if help is unavailable.

·         Lines 256-258 (rewritten with correction): permeability through the blood vessels. Only one study found that all tested compounds were inactive against hyaluronidase. The structures tested were xanthone derivatives with only methoxy, hydroxy or methylbromo moieties [54]. On the

·         Sections 2.1.4 requires additional references.

·         Section 2.2 needs many corrections for proper English usage and grammar. Several sentences are incomplete as written (e.g., see lines 296-299, lines 332-335 (nonsensical)).

·         Lines 357-359 (rewritten with correction): As mentioned above, xanthone derivatives demonstrate potent activity toward targets and pathways associated with vascular disorders. This activity warrants further investigation, specifically for potential usage of xanthone derivatives as active pharmaceutical ingredients in topical formulations.

The authors supplied several clarifications and corrections, but there are still significant issues with language, clarity, and structure accuracy in select graphics, as noted above. The authors are welcome to use my suggested corrected wording in the indicated lines but should also carefully revise and reference sections 2.2 and 2.1.4, respectively. Overall, the content and review emphasis is improved, but the delivery still falls short of publication quality in particular sections.

Comments on the Quality of English Language

I have offered wording suggestions for specific lines; however, indicated sections and longer passages still require more substantial editing by the authors.

Author Response

Dear Reviewer,

Kindly please see the attachments.

Thank you so much,

Authors

Round 3

Reviewer 2 Report

Comments and Suggestions for Authors

The authors made meaningful corrections as recommended from the second extensive review. Unfortunately, both the Scheme 1 drawing and the Figure 2 drawing are still incorrect and should not be published.

Hyaluronidase “cuts” the hyaluronic acid polymer, thereby creating two shorter polymers, each of which is terminated by a hydroxyl group—one at C-4 of the D-glucuronate and one at the anomer carbon (C-1) of the N-acetylglucosamine. The author’s graphic does not show this. It instead shows two new polymers but neither of which has a hydroxyl terminus. In addition, the new polymers drawn are impossible structures. See: https://proteopedia.org/wiki/index.php/Hyaluronidase for a helpful depiction that could be used as a template to create an accurate graphic.

The Figure 2 drawing is almost correct. The dashed bond must be behind the normal bond that it overlaps (select the dashed bond and use the ‘move to back’ or ‘move behind’ feature in your drawing program), while the carbonyl attached to the dashed bond should be moved downward vertically (just slightly) for clarity. As drawn, the dashed bond is not attached to a tetrahedral carbon atom.

Considering the authors’ implemented grammar corrections and general clarifications in specific sections, and with their added references, I recommend publication, however only after the indicated graphics are corrected and checked.

Author Response

Dear Reviewer,

Please see below our answers to your comments. We wanted to deeply thank you once again for the time and afford that you have put into reviewing our work.

"Hyaluronidase “cuts” the hyaluronic acid polymer, thereby creating two shorter polymers, each of which is terminated by a hydroxyl group—one at C-4 of the D-glucuronate and one at the anomer carbon (C-1) of the N-acetylglucosamine. The author’s graphic does not show this. It instead shows two new polymers but neither of which has a hydroxyl terminus. In addition, the new polymers drawn are impossible structures. See: https://proteopedia.org/wiki/index.php/Hyaluronidase for a helpful depiction that could be used as a template to create an accurate graphic."

Thank you for highlighting the mistakes that are still visible in the scheme. It was corrected and adjusted, also based on the link that you provided. Once again thank you so much for that.

"The Figure 2 drawing is almost correct. The dashed bond must be behind the normal bond that it overlaps (select the dashed bond and use the ‘move to back’ or ‘move behind’ feature in your drawing program), while the carbonyl attached to the dashed bond should be moved downward vertically (just slightly) for clarity. As drawn, the dashed bond is not attached to a tetrahedral carbon atom."

We corrected the figure, for the bond to be more clearly pictured and not cause misleading conclusions. Now the dash bond it is visibly attached to a tetrahedral carbon atom and dash bond it is behind the normal bond.

Hope made changes are now done correctly.

Sincerely,

Authors